Identification of fungi isolated from commercial bumblebee colonies

Chow Lui Julie 1
Nesbit Miles L. 1
Hill Tom 2
Tranter Christopher 2 3
Evison Sophie E.F. 4
Hughes William O.H. 5
Graystock Peter p.graystock@imperial.ac.uk pgraystock@imperial.ac.uk 1
1 Georgina Mace Centre for the Living Planet, Department of Life Sciences, Silwood Park Campus, Imperial College London , Ascot , Berkshire , United Kingdom
2 School of Biology, University of Leeds , Leeds , United Kingdom
3 School of Veterinary Science, University of Liverpool , Liverpool , United Kingdom
4 School of Life Sciences, University of Nottingham , Nottingham , United Kingdom
5 School of Life Sciences, University of Sussex , Brighton , United Kingdom
Colla Sheila
Electronic publication date: 2024 Jan 30
Publication date: 2024
Volume: 12
Electronic Location ID: e16713
Received 2023 Aug 14; Accepted 2023 Dec 4
Copyright: ©2024 Chow et al.
Copyright year: 2024
Copyright holder: Chow et al.
License: This is an open access article distributed under the terms of the Creative Commons Attribution License, which permits unrestricted use, distribution, reproduction and adaptation in any medium and for any purpose provided that it is properly attributed. For attribution, the original author(s), title, publication source (PeerJ) and either DOI or URL of the article must be cited.
License URL: https://creativecommons.org/licenses/by/4.0/

Keywords: Pollinator, Microbe, Fungus, Spillover, Bombus, Aspergillus, Candida, Penicillium, Zygosaccharomyces, Monascus

Funding: NERC Natural Environment Research Council, UK; NE/G012113/1 Bumblebee Conservation Trust This work was funded by the NERC (Natural Environment Research Council, UK; NE/G012113/1), and the Bumblebee Conservation Trust. The funders had no role in study design, data collection and analysis, decision to publish, or preparation of the manuscript.

==============================
Fungi can have important beneficial and detrimental effects on animals, yet our understanding of the diversity and function of most bee-associated fungi is poor. Over 2 million bumblebee colonies are traded globally every year, but the presence and transport of viable fungi within them is unknown. Here, we explored whether any culturable fungi could be isolated from commercial bumblebee nests. We collected samples of various substrates from within 14 bumblebee colonies, including the honey, honey cup wall, egg cup wall, and frass then placed them on agar and recorded any growth. Fungal morphotypes were then subcultured and their ITS region sequenced for identification. Overall, we cultured 11 fungal species from the various nest substrates. These included both pathogenic and non-pathogenic fungi, such as Aspergillus sp., Penicillium sp., and Candida sp. Our results provide the first insights into the diversity of viable fungal communities in commercial bumblebee nests. Further research is needed to determine if these fungi are unique to commercial colonies or prevalent in wild bumblebee nests, and crucially to determine the ecological and evolutionary implications of these fungi in host colonies.

Introduction

Pollinator-associated microbes are generally understudied unless they have been associated with the honey bee, Apis melliera (Yordanova et al., 2022). Since the 1990s growing commercialisation and international trade of other bee groups such as bumblebees (Bombus spp.) and some solitary bee species (e.g., Osmia cornifrons) has led to the need to understand what microbes are present in, and potentially globally transported with, these commercially reared bees (Graystock et al., 2016a; Graystock et al., 2016b; Graystock et al., 2020; Hedtke et al., 2015; LeCroy et al., 2022). In recent years the application of molecular screening and 16s sequencing has helped explore microbial diversity and movement patterns in commercially traded bees, but focus has largely remained on protists and bacteria (Huang et al., 2015; Milbrath et al., 2021). There remains a lack of appreciation of which fungal groups are present in commercially traded populations of bees, and even less of an idea of the risk they may pose to their hosts and to other bees.

Although there is a dearth of research describing bee-associated fungi, what evidence for we do have suggests that they can have a range of parasitic, mutualistic, or commensal relationships with the bees with which they associate (Evison & Jensen, 2018). The pathogenic fungi of honey bees are the best known because they associate with the most utilised agricultural bee species and their impact may be the most significant to human interests (Yordanova et al., 2022). Ascosphaera apis, for example, parasitises the developing larvae of A. mellifera (and some solitary bee species), killing the larvae leaving a chalky cadaver covered in transmissible and highly persistent spores, causing a disease in honey bees known as chalkbrood (Aronstein & Murray, 2010; Evison, 2015; LeCroy et al., 2022). Ascosphaera spp., along with other known fungal parasites pose a threat to bee health by impacting larval survival and leading to decline in reproductive success and (in the case of honey bees) collapse of entire colonies (Evison & Jensen, 2018; Lopes et al., 2015; Ravoet et al., 2014; Vojvodic et al., 2012).

Positive or neutral relationships between bees and fungi are likely to be common, because many bee-associated fungi can persist within the nesting materials and can sometimes grow on food provisions or frass rather than grow directly within the bee, but a general lack of understanding of the nature of their interactions make characterising mutualistic or commensal relationships difficult. An example of possible beneficial roles of fungi can be found with honey bees. There is strong evidence that mycelium of multiple polypore fungal species has a refractory effect on viruses such as deformed wing virus and Lake Sinai virus (Stamets et al., 2018). Several stingless bee species also gain significant benefits from fungi such as Scaptotigona depilis, Tetragona clavipes and Melipona flavolineata which cultivate a Zygosaccharomyces fungi that, at least in the case of S. depilis bees, must be consumed by larvae to initiate pupation (Menezes et al., 2015; Paludo et al., 2018; Vit, Pedro & Roubik, 2013). If the Zygosaccharomyces fungi also benefit from this relationship—via dispersal and the provision of habitat—then perhaps these are examples of mutualism, but this is not yet known. In addition, some fungal species could include roles aiding digestion and parasite defence (Stefanini, 2018). There is a clear need to improve our understanding of the species and interaction diversity of fungi and bees, only then can we start to build a comprehensive understanding of the potential risks or benefits fungi pose to bee health.

It is estimated that globally over two million bumblebee colonies are purchased annually for agriculture (Graystock et al., 2016a; Graystock et al., 2016b). While these colonies presumably contain favourable substrates for harbouring or cultivating fungi, they are often transported internationally with little to no appreciation of what they might contain (Dharampal et al., 2020). The adult bumblebees in commercial nests, and the pollen they are fed, have occasionally been screened for certain pathogenic fungi, namely Nosema spp. (now recategorized as Vairimorpha) and Ascosphaera spp., which have been detected (Graystock et al., 2013a; Graystock et al., 2013b). However, in certain countries such as the UK, such screenings are not a prerequisite for trade or distribution and are only performed at the supplier’s discretion (Cordes et al., 2012; Dharampal et al., 2020; Graystock et al., 2016a; Graystock et al., 2016b; Graystock et al., 2013a; Graystock et al., 2013b; Graystock, Goulson & Hughes, 2015; McIvor & Malone, 1995). International trade of bumblebee colonies for agricultural use is known to have facilitated the spread of parasites and viruses between managed and wild pollinators (Colla et al., 2006; Goka et al., 2000; Graystock et al., 2016a; Graystock et al., 2016b; Graystock, Goulson & Hughes, 2015; Graystock et al., 2020; Sachman-Ruiz, Narváez-Padilla & Reynaud, 2015). The focus on honey bee parasites has, however, left us unaware of the wider variety of microbes, specifically fungi that may be present in commercial bumblebee colonies.

Within a typical bumblebee colony there are several different substrates that could culture or harbour fungi. Of particular interest are bumblebee honey, frass, and nest wax associated with eggs and honey cups (we differentiate egg cups and brood here because we are referring to only the wax around the brood, and not the brood itself) (Fig. 1). Though honey from bumblebees is infrequently studied, honey bee honey can exhibit antibiotic properties, and is also known to harbour growth of xerophilic and xerotolerant fungal species, such as Ascosphaera apis, Bettsia alvei and species of Alternaria, Aspergillus and Penicillium (Kačániová et al., 2009; Kačániová et al., 2012; Rodríguez-Andrade et al., 2019; Saksinchai et al., 2012; Snowdon & Cliver, 1996). Bumblebee honey is not capped and is not stored long term, so it has greater similarity to nectar. Nectar has a lesser-known diversity of fungal genera, with the most common being Metschnikowia and Aureobasidium (De Vega, Herrera & Johnson, 2009; Jacquemyn et al., 2013; Mueller, Francis & Vannette, 2023; Pozo, Herrera & Bazaga, 2011). Bumblebee nests (particularly in commercial settings) have frass areas where bee faeces and waste (including dead adults and larvae) accumulate. Across several bee species, frass has been associated with bee pathogenic fungi, such as Ascosphaera (honey bees) and Nosema (honey bees and bumblebees) and plant pathogenic fungi such as Botrytis cinerea and Colletotrichum acutatum (Inglis, Sigler & Goette, 1993; Parish et al., 2019; Wynns, Jensen & Eilenberg, 2013). Bumblebee wax is used to house developing larvae and store honey. Whilst there are no data on fungi found in bumblebee wax, studies linking Zygosaccharomyces sp. (originally Monascus sp.) as an obligate mutualist in stingless bee brood development (Menezes et al., 2015; Paludo et al., 2018), combined with the known antimicrobial properties of honey bee wax (Grange & Davey, 1990), provide an interesting juxtaposition of predicted habitability of this substrate.

Figure 1 Image of a Bombus terrestris audax nest taken at Silwood Park, Ascot, United Kingdom.

It includes magnified pictures of an egg cup (left), a honey cup (centre left), honey (centre right), and frass (right). Photo credit: Miles L. Nesbit.

Despite well researched information on the presence, functional roles, and impacts of some honey bee fungi, relatively little is known about the fungi associated with bumblebees or their nest substrates. In addition, most of this research has focused on fungal DNA, so the viability of any fungi identified remains unknown (Jacquemyn et al., 2013; Mueller, Francis & Vannette, 2023; Pozo, Herrera & Bazaga, 2011; Wojcik et al., 2008). There is a clear gap in knowledge regarding the community of viable fungal microbes associated with bumblebee nests, which provide multi-component environments for brood development, food storage and debris accumulation (Grabowski & Klein, 2017). Here we survey and ID the fungi present and culturable in commercially reared bumblebee nests, allowing for the first assessment of the potential for the distribution of these fungi within commercially sourced bumblebee nests.

Materials and Methods

Tissue collection

A total of 14 commercially produced Bombus terrestris audax (Linnaeus, 1758) colonies were purchased from Biobest NV (Westerlo, Belgium) in 2012. Immediately upon arrival to the laboratory in Leeds (United Kingdom) from continental Europe, all adult bees, which included around 80–100 workers and a queen per colony, were removed to allow careful and sterile collection of four different colony substrates that were identified as likely microbial substrates (Fig. 1). The substrates collected were bumblebee honey (H), wax from egg cups (ECW), wax from honey cups (HCW) and frass from the outer edge of the colony (F) (Fig. 1). From each colony, approximately 1–2 cm3 of each wax/frass material were collected using flame-sterilised forceps and 0.5 ml of honey samples were collected with a pipette. The samples were placed in quadrants of a Petri dish that contained solid malt extract agar (MEA) and 0.01% streptomycin, to favour the growth of moulds, yeasts and fungi, over bacteria. The Petri dishes were incubated at 30 °C for 6 days after which point microbial growth was clearly detectable across plates and substrates. During incubation microbial growth was monitored with daily photographs.

After the 6-day period, sub-cultures of morphologically distinct fungal growth were looped and isolated. In total, a series of three sub-cultures were performed for each isolate to ensure it was the only strain growing on the plate prior to molecular identification. A representative sample of each morphotype growing from each substrate was sequenced. To extract the fungal DNA, approximately 0.05 g of the fungus was scraped off the media and added to 200 µL of 5% Chelex suspension (in 10 mM Tris buffer) and 0.05 g of 0.1 mm Zircona/Silicabeads, and placed in a Qiagen Tissue Lyser (Qiagen, Hilden, Germany) for 4 min at 50 oscillations/s. Samples were then incubated in a 90 °C water bath for 20 min and then centrifuged for 30 min at 8 °C. The supernatant was cleaned with OneStep-96 PCR Inhibitor Removal Kit (Zymo Research, Irvine, CA, USA) prior to PCR.

Fungal identification

PCR was performed using fungal primers that target conserved regions of the 18S (ITS 1) and the 28S (ITS 4) rRNA genes to amplify the intervening 5.8S gene and the ITS 1 and ITS two noncoding regions (Henry, Iwen & Hinrichs, 2000; White et al., 1990; Table S1). The PCR products were then sent to Eurofins for Sanger sequencing. Fungal identity was determined by NCBI BLASTn searches using fungi as the search organism and strictly highly similar sequences (megablast) as parameters. For multiple identical hits, the potential species were listed (Supplemental Information—Table S1) and the species with the largest number of matching isolates were chosen. To show the relationships between the identified species, Mega11 (Kumar et al., 2018) was used to create a phylogenetic diagram based on NCBI taxonomy (Letunic, Doerks & Bork, 2015).

Statistical analysis and quantifying fungal growth

In each of the substrates, the broad microbial growth (not specific to fungal species) was quantified from photographs of the media plate using ImageJ. This method allowed us to record microbial growth in general, but it could not provide a breakdown of growth by particular fungal species. The areas of growth per colony/substrate were summed and converted into binary presence-absence of microbial growth data. If there was any growth across the six-day period, the substrate was considered to have a viable fungus. If there was not, the growth was considered absent. A binomial GLM was run to determine the effects of the different substrates and colonies on the presence or absence of growth (McCulloch & Rossi, 1994). To assess the significance of predictors collectively, the Anova function from the car package was employed (Fox & Weisberg, 2019), utilizing Type II sums of squares. This approach facilitates the extraction of p-values associated with each predictor, thereby contributing to the evaluation of the overall model fit. The presence of specific species in a substrate was confirmed with Sanger sequencing.

Results

Substrate type is associated with different fungal growth

Frass exhibited the lowest incidence of viable fungi, with only 35.7% (six) of the 17 samples containing such fungi. However, this substrate showcased the highest diversity of fungal species. A greater percentage (71.4%) of the honey cup wall and the egg cup wall samples had viable fungi, and 85.7% of honey samples had viable fungi (Fig. 2). Substrate type had a statistically significant effect on the occurrence of viable fungi (X2 = 14.134, df = 3, p = 0.0027); there was also significant difference in presence or absence of growth between bumblebee colonies (X2 = 27.15, df = 13, p = 0.0119).

Figure 2 Evidence of fungi growing from various bumblebee nest substrates.

The presence (blue) and absence (red) of viable fungi in the four different substrates from bumblebee nests.

Multiple species of fungi were cultured from bumblebee nest substrates

A total of 11 species across five genera were isolated from the morphotypes collected from the four substrate types. All the identified fungi belonged to the phylum of Ascomycota, represented by classes Saccharomyces and Eurotiomycetes. The three species from the class Saccharomyces were the yeasts Zygosaccharomyces rouxii, Candida orthopsilosis and Candida parapsilosis. The eight fungal species from the class Eurotiomycetes were Penicillium sp., five species of the mould Aspergillus, and one of Monascus ruber. The isolates of Aspergillus were classified as A. tubingensis, A. puniceus, A. niger, A. fumigatus, A. flavus. The isolates of Penicillium were identified as P. citrinum and P. crustosum. Whilst the culture media is known to be general to many moulds and yeasts, we cannot rule out the presence of other fungi that were not culturable in this particular media. These results are therefore likely to be an underestimate of total fungal species present.

A high degree of certainty (>99% confidence with >98% coverage and top 100 matches in agreement) can be attributed to the identifications of Penicillium crustosum, Zygosaccharomyces rouxii, Aspergillus fumigatus, Candida orthopsilosis, Candida parapsilosis (74% coverage with 100% confidence), Penicillium citrinum, Aspergillus niger, and Aspergillus puniceus. The identification of Aspergillus flavus, Aspergillus tubingensis, and Monascus ruber were less certain; they were determined to be the likeliest candidate species for the isolate (>99% confidence with >98% coverage but had multiple species with perfect matches; Supporting Information–Table S1). In addition, because we only sequenced a single representative of each morphotype, it is conceivable that additional species may be present which were not morphologically distinct from those identified.

Various substrates harboured viable fungi

The highest diversity of fungi was cultured from frass (Fig. 3), where six out of 11 isolates included both Saccharomyces yeasts and Eurotiomycetes fungi. A high richness of isolates (five), were also obtained from the honey cup wax. Egg cup wall contained three fungal species and only a single isolate was cultured from the honey (Fig. 3). The order of species richness from greatest to least was therefore (1) frass; (2) honey cup wall; (3) egg cup wall; and (4) honey. Two of the Aspergillus species (A. puniceus and A. flavus) were found only in the egg cup wall. We found A. tubingensis in the honey exclusively. In frass we found A. niger, C. orthopsilosis, and P. citrinum. We found P. crustosum in the honey cup wall. In both frass and the honey cup wall we were able to isolate A. fumigatus, Z. rouxii, and C. parapsilosis. We found M. ruber in both the honey cup wall and the egg cup wall (Fig. 3).

Discussion

This study detected 11 different fungal species from four substrates (frass, honey, honey cup wax, and egg cup wax) within commercial colonies of the bumblebee Bombus terrestris audax. This is the first survey of fungal diversity grown from different substrates of commercially reared bumblebees, and provides some interesting insights into how and why such diversity might differ.

Frass is the detritus from bumblebee activity in the nest. It is often a mixture of chewing debris and excrement. The highest fungal diversity was observed in frass, which can be attributed to its role within the colony. Frass has exposure to bee excretion and a number of phoretic mites which all of which may contribute to its high fungal diversity (Osimani et al., 2018; Paniagua Voirol et al., 2018; Pernice, Simpson & Ponton, 2014; Schwarz & Huck, 1997; Weiss, 2006). Frass may hold the highest diversity as well because of the length of time the fungi have to grow and unhomogenized nature of the substrate. There is also the consideration that there is significantly less diversity and abundance of microbes in insects than in (for example) mammals due to the physical barriers between the gut and the rest of the insects’ systems such as the basal lamina (Pernice, Simpson & Ponton, 2014). Almost every phylogenetic grouping found in the colonies was represented in frass (with the exception of Monascus ruber). Candida orthopsilosis and Candida parapsilosis were found in the frass. The isolation of Candida strains on bee surfaces suggests that these yeasts rely on insect vectors for dispersion across flowers but have no negative effect on the bees (Hong et al., 2003). Yeasts are often found in the nectar of flowers visited by bumblebees (Herrera et al., 2009), potentially showing that bumblebees contain yeasts prior to contact with nectar from their preferential flowers, although this could still be an effect of the rearing process. There is also evidence that yeasts use flowers as sites for mass reproduction and insects as platforms for hibernation (Brysch-Herzberg, 2004).

Figure 3 The identity and location various fungal species were isolated from within a bumblebee nest.

The fungal isolates cultured in this study from the various substrates. The star represents the genus Aspergillus, the hexagon represents the genus Monascus, the circle represents the genus Penicillium, the diamond represents the genus Zygosaccharomyces, and the square represents the genus Candida. These species are related in a diagrammatic phylogeny. Mold species are represented in brown and yeast species are represented in blue. Each species is listed as either present or absent in the four substrates. The location and totals are summarized in the chart. The substrate column includes a small representative image of the substrate the fungal isolate was found in.

Bumblebee larvae are fed with honey produced by the queen bumblebee and her workers. Honey had the lowest diversity of fungal isolates, which could be attributed to its antimicrobial properties (Aurongzeb & Azim, 2011; Nolan, Harrison & Cox, 2019; Wahdan, 1998). However, there is little evidence to suggest that bumblebee honey, which is significantly closer to nectar in terms of water content than honey bee honey, has similar antimicrobial effects. The isolate found in the bumblebee honey, Aspergillus tubingensis, is a member of the Aspergillus genus which have long been known to be associated with honey bees and includes the pathogen responsible for the highly infectious stonebrood disease in immunocompromised bees (Cheng et al., 2022; Foley et al., 2014). Bumblebee honey is stored in honey cups for future feeding to larvae. The honey cup walls had the second-largest variety of fungal isolates. One interesting fungal isolate was Penicillium crustosum, which causes food spoilage (De Jesus et al., 1983) and has not traditionally been associated with bumblebee species. It is unclear if this fungi spoils the nutritional value of bumblebee honey but if it does, it would be a harmful microbe to colony health even if it does not parasitize the bees themselves.

Bumblebee eggs are placed in small clusters within egg cups where they are nursed and fed by worker bees before being moved to individual cups prior to pupation. The egg cups contained two fungi known to be pathogenic in honey bees but with unknown affects in bumblebees (as of yet—A. tubingensis and A. niger) and one that has been associated with improved nutrition and pupation in stingless bee species (Monascus ruber). Monascus ruber was originally found to be crucial for the survival of a Brazilian stingless bee, Scaptotrigona depilis, by being a food source for its larvae, though this was later corrected by Paludo et al. (2018) to Zygosaccharomyces sp. (Menezes et al., 2015; Paludo et al., 2018), growing inside brood cups over the larval food and being transmitted across generations via recycled contaminated nest materials. Our study identified M. ruber by the same PCR primers (ITS-1 and 4) so it very well could be misidentified similarly—this was later identified in Paludo et al. (2018) with species specific primers. There is also evidence of M. ruber in the brood cups of stingless bees, although it may not serve the same purpose.

All sampling was done from colonies sourced from only one commercial supplier of bumblebees and with no similar data from wild or other suppliers, no comparisons can be made. Nevertheless, the presence of live and culturable fungi confirms commercially imported bumblebee colonies as a source of fungal microorganisms. However, if we are to fully understand the frequency and, indeed, the full diversity of such fungi, more sampling is required. This study provides a snapshot of the fungal profiles within bumble bee colonies ordered from one supplier at one point in time. Suppliers could employ various pollen/facility sterilization techniques and/or rearing methods to reduce or remove the fungi present. Commercial colonies also are not representative of wildtype colonies and likely contain different diversities of microorganisms (Newbold et al., 2015). There is likely spillover of nest fungi between commercial and wild nests, similar to the pathogenic spillover documented in the UK, the US, Ireland, and Japan (Cameron et al., 2016; Graystock et al., 2016a; Graystock et al., 2016b; Graystock, Goulson & Hughes, 2015; Newbold et al., 2015; Niwa et al., 2004), but to fully understand the risks this may or may not pose to wild bumblebees, more research is required. Because the source colonies from this study were isolated and maintained indoors, having never been free-flying in the field, the likely entry points for the fungi found would be either from the founding queens that hibernated in soil or peat materials or through contamination of rearing provisions, such as pollen (David et al., 2015; Dharampal et al., 2020; Mommaerts, Jans & Smagghe, 2010). Although commercial suppliers now often sterilize the pollen that they feed to colonies, the sterilization methods employed have inconsistent effects on fungal spores (Strange et al., 2023), and the overall effectiveness of sterilization against a wide range of fungal spores is unknown (Graystock et al., 2016a; Graystock et al., 2016b).

Conclusions

Though we identify a diversity of fungi from the various substrates of commercial bumblebee colonies, we do not yet know the functional significance of these fungi or the generality of the findings. Further research is needed to explore the interactions between bumblebees and fungi, including the ecological and evolutionary implications of these interactions. This research can inform conservation efforts and help to ensure the continued pollination services provided by commercial and wild bumblebees, which are essential for maintaining ecosystem health and food security.

Supplemental Information

Table S1 Sequencing data and metadata

Summary of the sample metadata associated with each sequence obtained in this study

Click here for additional data file.

Table S2 Fungal growth data

Raw data on the presence of growth from the various substrates over time.

Click here for additional data file.

Supplemental Information 3 R Code used to analyse growth data

Click here for additional data file.

Additional Information and Declarations

Competing Interests

Author Contributions

DNA Deposition

Data Availability

The authors declare there are no competing interests.

Lui Julie Chow analyzed the data, authored or reviewed drafts of the article, and approved the final draft.

Miles L. Nesbit analyzed the data, prepared figures and/or tables, authored or reviewed drafts of the article, and approved the final draft.

Tom Hill performed the experiments, authored or reviewed drafts of the article, and approved the final draft.

Christopher Tranter performed the experiments, authored or reviewed drafts of the article, and approved the final draft.

Sophie E.F. Evison conceived and designed the experiments, authored or reviewed drafts of the article, and approved the final draft.

William O.H. Hughes conceived and designed the experiments, authored or reviewed drafts of the article, and approved the final draft.

Peter Graystock conceived and designed the experiments, performed the experiments, authored or reviewed drafts of the article, and approved the final draft.

The following information was supplied regarding the deposition of DNA sequences:

The sequencing data is available at GenBank: OR287164–OR287179.

The following information was supplied regarding data availability:

The sequencing and sample metadata and the growth data and analysis code are available in the Supplemental Files.

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
