# Peer review of "Identification of fungi isolated from commercial bumblebee colonies"

_PeerJ, doi:10.7717/peerj.16713_

## Round 0.1 · original submission · Minor Revisions

We have received comments from three reviewers which I believe would largely strengthen the manuscript. I hope you are able to address them and I look forward to receiving the revised manuscript.

Reviewer 1 ·

Basic reporting

The article is overall clear and easy to follow, but certain sections require rewording to improve clarity and accuracy:

L65 – Please change “Varimorpha” to “Vairimorpha”

L87-88 – I don’t believe Metschnikowia is mentioned in Menezes 2015 and Paludo 2018, are you referring to the Monascus species isolated from stingless bee cerumen in these papers?

L75, 106 – Please use consistent formatting when referring to figures in the text (figure 1 vs. fig 1)

L116 – should this be 200uL?

L146 – 85.7% is listed twice, please remove one occurrence

L205-206 – This was corrected in a future publication, Paludo et al. 2018 Scientific Reports, which identified the fungus eaten by bee larvae as Zygosaccharomyces. Please change this discussion point to reflect this
.
Supplementary table 2: Fun_growth.csv – Please include a readme file with information on each column, as was provided for Supplementary table 1.

Experimental design

The experimental design is appropriate for the question being asked, I have just one comment for improvement:

L128 – Choosing fungal identity based on known bee connections seems biased, please justify these assumptions further or edit discussion of fungal isolates to include best match according to sequencing, even if these are genus-level annotations.

Validity of the findings

This manuscript fills an important knowledge gap on viable fungal communities isolated from various substrates within commercial bumble bee nests. Overall, I think the results are valid, and will be useful knowledge in the field of bee-microbe interactions. I have one concern about the presentation of results:

L113-114 – It is not clear to me from the methods whether each unique morphotype for each sample was sequenced, or if all samples were considered together when choosing morphotypes. Please clarify. If the second method was used, please also specify in your results section that the diversity results you are presenting on based on morphotypes, and not necessarily on species, as these are not always analogous.

Reviewer 2 ·

Basic reporting

56-57: Fungi can also aid in the reduction of pathogens in honey bees. See Stamets et al 2018, https://www.nature.com/articles/s41598-018-32194-8

141-146: Switching the order of this paragraph would make the results easier to follow. Start with the proportions then talk about the statistical significance (or lack thereof).

226-228: see Strange et al 2023 (https://doi.org/10.1093/jee/toad124) for pollen sterilization techniques and their efficacy against some known fungal bee pathogens.

Experimental design

105: Is there an approx. size of the colonies or number of workers in each colony that could give the reader an idea of colony maturity/size vs. fungal profile?

108: Was only a single sample of each substrate taken from a colony? Worded a different way: How many replicates of each substrate were taken from a single colony?

111-112: Why was 6 days selected for the growth period?

128: What is meant by “simply more common”?

132-139: It's not clear why images were taken or how they were analyzed. What exactly signified “Yes, X fungus was present in Y substrate”? Was it seeing the culture in the dish? Or was it something revealed during analysis using ImageJ?

Validity of the findings

144: Is this saying that frass contained more fungal diversity, but the spores isolated from frass did not germinate as often as frass isolated from other substrates? How many frass samples were taken and how many did not germinate?

145: 35.7% of how many samples?

167-168: this is great, think about including the same but for spore viability.

187-189: Expand on why you think a commensal relationship is likely?

212-213: This study provides a snapshot of the fungal profiles within bumble bee colonies ordered from one supplier at one point in time. While full colony sterilization is not possible, suppliers could employ various pollen/facility sterilization techniques and/or rearing methods that could have an effect on the fungi present.

Additional comments

The introduction was clear, concise, and thorough, providing information that sets the stage well for the following. The research objective was well-stated and justified.

There are several run-on sentences throughout the manuscript (e.g., 63-68; 159-162; 208-211).

Typically, bumblebee domiciles are referred to as nests, not hives (used interchangeably throughout the manuscript). Additionally, egg cups are typically referred to as brood. I recommend changing throughout for consistency’s sake with respect to the literature.

143: delete “in this”.

155: delete “and” at the beginning of the sentence.

173: Referencing Figure 3 more throughout this paragraph would be useful

202-203: delete “each”, insert “to” before pupation.

218-222: This reads as two sentences saying essentially the same thing.

231: but you applied generalities to 212-213.

Why do you think A. niger and P. citrinum were found only in the frass? How do you think it moves through the colony?

The conclusions could benefit from a closing statement regarding the novelty of the work and some broad summary sentences.

·

Basic reporting

Several line by line comments below center around the issue of clarity. In the beginning of the manuscript the authors lay out an argument that, although information is known about microbes affecting honey bees, most other bee species have a lack of data regarding fungal pathogens, symbionts or commensals. I believe there are several areas in the manuscript where, although presenting this dichotomy early in the manuscript, the authors use honey bee generated data as support and other areas where the authors use solitary or non-honey bee data as support. I would urge the authors to use the term “honey bee” when appropriate and be as clear as possible otherwise so the reader can understand the points.

Line 30: The term “largely ignored” is too subjective to describe the bulk of research on non-honey bee microbe research and I feel is inaccurate to support with your reference (Yordanova et al. 2022). In this reference the authors argue that the bulk of data is disproportionate with the diversity of bee species, but do not argue that research on other bee species is ignored, just poorly represented. I would eliminate the term “largely ignored” from this sentence or replace with a less subjective evaluation of the research.
35 Clarify what “this” is- do you mean what microbes are present in general?
36 Viruses have no ribosomes and cannot be sequenced for the 16S region- alter the sentence or clarify here.
38 In this case viable functionally means that the fungus is able to reproduce and grow- the "genuinely alive" should be removed.
39 unclear what “Awareness” refers to here- previously you stated that microbe studies with honey bees are plentiful, do you mean fungi associated specifically with bees other than honey bees? Also, suggest change of awareness to something less subjective such as “research describing,” or provide a metric that measures “awareness.”
46 Here you refer to “bee health” and “colonies” supposedly meaning honey bee? I think with your effort here to study other bees, best to be clear here when you make statements in the sentence. Plus, stating colony would be inappropriate for solitary bee species.
46 Can you elaborate on what aspect of life history makes study difficult, or provide a reference? I am assuming you mean difficulty in culturing symbiotic fungi in vitro?
63 Does this refer to adult bees in the commercial nests, or are there separate commercial and non commercial adults.
64 Nosema was reclassified as Varimorpha, not renamed as. The genus already existed.
72 Here you say “particularly fungi”, but your two examples for screening in Bombus hives were fungi. I would eliminate this part of the sentence or give more non-fungi examples prior to this statement.
170 “in the honey alone,” does this mean that it was only found in the honey, or was that the only species found in the honey? Or both? Clarify.
180-181 Do you have a reference to support this definition? I am only aware of the term frass being used for the excrement of insects. Does this definition mean that the chewing debris and insect body parts have gone through the digestive tract? I think I your study you have defined a “frass area” which is the perimeter of the next where frass and this other material is deposited. Please clarify or provide a reference to support your term here.
182 I’m not sure what you mean by “bumblebee systems”. Internal systems like digestion and excretion? Multiple colonies? Commercial rearing systems?
189 Isolation found in your study or in bee surfaces in general? Clarify.
203-204 clarify that by potentially pathogenic you mean it’s a pathogen in honey bees but no known confirmation of pathogenicity in bumblebees.
223 What do you mean by isolated indoors? Do you mean that gynes produced by another indoor colony were used to establish the one you investigated? Clarify.
254 and 336 This journal is abbreviated but most of the others are not. Maintain similar style.
415 Geographic info for Silwood park is needed. City and country or similar.
415 “zoomed in” is more slang in usage here. Change to something along the lines of magnified or detailed and include scale bars in figure.
420-426 Here the scientific names need to be either underlined or non italicized.

Experimental design

By nature, fungi that will be living in xerophilic environments with a strong osmotic potential would need a similar media in which to isolate them from their substrates. In addition, gut pathogens of many bees may need a high CO2 level to germinate or a specialized media in which to grow. While the use of MEA is reasonable for the growth of many molds and yeasts, how do the researchers know that they have fully sampled the microbes in the bumblebee environment without using more than one media source or culture protocol? The presence or absence of a species in your results may be due more to the culture technique than the actual presence. In addition, I need more information regarding the establishment and maintenance of the colony before laboratory analysis. How do we know the presence of the yeast is supported by the biology of the bees and not from the contamination of the sugar syrup the bees are presumably fed as part of the rearing and shipping? The presence of mold species can also be due to contamination of variable other rearing supplies. It would be reassuring to test these components individually as well

139 Can you tell us why the Chi Sq. was performed? As a multiple comparison test between groups after the analysis?

199-200 Do we have evidence that this fungal species was not present in the sugar syrup given to the colony by the company?

Validity of the findings

the authors need to more clearly relay the limitations of the experimental design (using only one media source and culture technique) to the readers These limitations greatly reduce the claims that can be made in the conclusion, as some of the fungi bee relationships proposed have no scientific basis for these comparisons other than the detection, which may be from anthropomorphic sources.

188 There is no evidence necessarily of commensal interaction with the data you present here. How do we know that the yeast didn’t grow on the frass after deposition or if the yeast was present in the sugar syrup shipped with the colony? Need to restructure this part to clearly show the limit of your study but suggest that some yeasts may exist as part of a commensal relationship.

192-193 I do not think there is sufficient evidence in this statement that the bumblebee honey has antimicrobial properties, as the references you present are honey bee studies and in your manuscript in lines 75-81 you lay out the premise that bumblebees may produce something between honey and nectar. Since honey creates a xerophilic environment, how do you know that the media you selected is osmotically sufficient to culture the fungi that grow in this environment? You need to provide a bumblebee specific reference here to support your antimicrobial claim, and mention the limitations of your culture method.

213 with the ubiquitous nature of Aspergillus contaminations in microbial laboratories, I would argue that there could be supplier specific contamination of the sugar syrup, plastic materials and pollen provision. Also it is unclear what you mean by colony sterilization here. I would clarify to explicitly say what you mean here (wild caught bees/ unable to sterilize plastic ware/ unable to sterilize colonies once they are established)

Additional comments

I agree that this is an important study to do and agree that further comparison with other cultured or wild bees would be quite interesting!

---

## Round 0.2 · accepted · Accept

Many thanks for clearly addressing the reviewer comments. I am happy to accept your manuscript for publication.